# Stereotactic MR-Guided Radiotherapy for Adrenal Gland Metastases: First Clinical Results

**DOI:** 10.3390/jcm12010291

**Published:** 2022-12-30

**Authors:** Morgan Michalet, Ons Bettaïeb, Samia Khalfi, Asma Ghorbel, Simon Valdenaire, Pierre Debuire, Norbert Aillères, Roxana Draghici, Mailys De Méric De Bellefon, Marie Charissoux, Pierre Boisselier, Sylvain Demontoy, Alexis Marguerit, Morgane Cabaillé, Marie Cantaloube, Aïcha Keskes, Touria Bouhafa, Marie-Pierre Farcy-Jacquet, Pascal Fenoglietto, David Azria, Olivier Riou

**Affiliations:** 1Montpellier Cancer Institute, Federation of Radiation Oncology of Mediterranean Occitanie, University Montpellier, INSERM U1194 IRCM, 34298 Montpellier, France; 2CHU Hassan II, Radiotherapy and Brachyterapy, 30050 Fez, Morocco; 3Institut de Cancérologie du Gard, University Federation of Radiation Oncology of Mediterranean Occitanie, CHU Carémeau, 30900 Nîmes, France

**Keywords:** magnetic resonance-guided radiotherapy (MRgRT), adrenal gland metastases (AGM), stereotactic body radiotherapy (SBRT), adaptive radiotherapy

## Abstract

Stereotactic MR-guided Radiotherapy (MRgRT) is an interesting treatment option for adrenal gland metastases (AGM). We reviewed data from 12 consecutive patients treated with MRgRT for an AGM in our center between 14 November 2019 and 17 August 2021. Endpoints were tolerance assessment, the impact of adaptive treatment on target volume coverage and organs at risk (OAR) sparing, local control (LC), and overall survival (OS). The majority of patients were oligometastatic (58.3%), with 6 right AGM, 5 left AGM and 1 left and right AGM. The prescribed dose was 35 to 50 Gy in 3 to 5 fractions. The median PTV V95% on the initial plan was 95.74%. The median V95% of the PTVoptimized (PTVopt) on the initial plan was 95.26%. Thirty-eight (69%) fractions were adapted. The PTV coverage was significantly improved for adapted plans compared to predicted plans (median PTV V95% increased from 89.85% to 91.17%, *p* = 0.0478). The plan adaptation also significantly reduced Dmax for the stomach and small intestine. The treatment was well tolerated with no grade > 2 toxicities. With a median follow-up of 15.5 months, the 1–year LC and OS rate were 100% and 91.7%. Six patients (50%) presented a metastatic progression, and one patient (8.3%) died of metastatic evolution during the follow-up. Adaptation of the treatment plan improved the overall dosimetric quality of MRI-guided radiotherapy. A longer follow-up is required to assess late toxicities and clinical results.

## 1. Introduction 

The adrenal glands are the fourth metastatic dissemination site of solid tumors after the lungs, liver, and bones. The prevalence of adrenal gland metastases discovered on post-mortem examination is up to 3.1% and even up to 38% for patients with malignancies [1,2]. The usual primary cancers involved are, in order of frequency: melanomas (almost 50%), lung cancers, breast cancers, kidney cancers and gastrointestinal cancers [2,3].

They are deep, asymmetric retroperitoneal pair organs, in close contact with the liver, kidneys, and large vessels such as the aorta or inferior vena cava [4]. This anatomical situation is a therapeutic challenge for surgeons and radiation oncologists, with possible severe treatment complications.

In settings of oligometastatic cancers, the local treatment of all metastatic sites showed promising results with increased overall survival in various cancers [5,6]. For adrenal gland metastases, surgery was often considered the standard treatment, providing a mean local control at 2 years of 84% [7]. For inoperable patients or those declining surgery, stereotactic body radiotherapy (SBRT) represents a good therapeutic alternative with clinical results similar to surgery (2 years local control of 84.8%) [8]. With the recent published excellent results on SBRT in oligometastatic setting, the technique can even be now considered as a standard treatment option [9]. This innovative technique allows dose escalation to target volumes (BED doses up to 100 Gy) with limited doses to adjacent organs at risk (OAR), thanks to a very strong dose gradient [7,10,11].

Magnetic resonance-guided radiotherapy (MRgRT) is a new technique of radiotherapy combining a linear accelerator with magnetic resonance imaging [12]. This technique allows for a better delineation of abdominal organs thanks to good soft tissue contrast of MRI, the possibility of daily dosimetric adaptation thanks to an integrated treatment planning system (TPS), and the possibility of gating with continuous cine-MRI acquisition. MRIdian^®^ is a device combining a 6 MV linear accelerator with a 0.35 Tesla MRI. It is particularly adapted for adrenal gland stereotactic radiotherapy, ensuring optimum precision in the delineation of target volumes and organs at risk, as well as daily adaptation of the plan to the changes in anatomy [13,14,15].

The aim of this prospective registry study was to report our initial experience on the treatment of adrenal gland metastases (AGM) with MRgRT. The first objective was to evaluate treatment tolerance. The secondary objectives were to evaluate the dosimetric benefits of adaptation and the clinical results. To our knowledge, this is the first study to evaluate toxicities and outcomes in patients treated with MRgRT for AGM.

## 2. Materials and Methods

### 2.1. Patient Selection

All patients treated with MRgRT for an adrenal gland metastasis at the Montpellier Cancer Institute from 14 November 2019 to 17 August 2021 were included. All patients had an assessment of metastatic disease with a CT-scan and/or PET/CT scan as well as a biological assessment. The inclusion criteria were: Eastern Cooperative Oncology Group (ECOG) performance status score ≤ 2, no MRI contraindications (presence of non-MRI compatible implanted cardiac devices, claustrophobia, psychiatric disorders, and metal objects), and a controlled metastatic disease by systemic treatment. Systemic treatments for primary cancer and metastatic disease were stopped (unless immune checkpoint inhibitors) during the radiotherapy. The indication of MRgRT had to be validated in a multidisciplinary tumor board. The study was registered in the Health Data hub (registration number: #1802) and was approved by our local research committee (2020/01). All patients signed an informed consent form before treatment.

### 2.2. Treatment Preparation

The procedure was similar to our pancreatic tumor experience [16]. All patients underwent a CT simulation directly followed by a 0.35 T MRI simulation using the MRIdian^®^ (Viewray, Oakwood Village, OH, USA) to ensure the reproducibility of the anatomic configuration. MR and CT images were rigidly registered for target volume delineation. Only the MR images were used for OAR delineation. A 1.5-T MRI simulation in our radiology department was also required to allow for better gross tumor volume (GTV) delineation after registration with MRIdian^®^ images. The simulation exams could be performed with contrast agents if indicated and no patient contraindications. Patients could be asked to fast for at least 3 to 4 h prior to all simulation exams (and every fraction) in case of close contact with the digestive OAR. Patients were in a supine position with arms down at their sides, and immobilization was obtained with a Totim^®^ (Essebi Medical, Faetano, Italy) device. Furthermore, for dose calculation, CT to MR image registration was performed using an elastic registration algorithm. During the CT simulation, MRI dummy surface coils with similar electron attenuation properties to real MRI coils were placed on the custom immobilization device. MR images were acquired with true fast imaging with steady-state free precession (TRUFISP) sequences (T1/T2 weighted, breath-hold technique (physiologic end-expiration), 17 to 25 s, 1.6 × 1.6 × 3 mm or 1.5 × 1.5 × 3 mm resolution, 45 × 45 × 24 to 54 × 47 × 43 maximum field of view).

### 2.3. Breath-Hold Procedure

All patients were simulated and treated with a breath-hold technique. They received a document explaining the respiratory breath-hold procedure and the terms that were going to be used during simulation and treatment. Patients were asked to perform respiratory breath-hold exercises at home. Another respiratory coaching session was performed directly before the first simulation. Breath-hold was achieved by voice guidance by radiotherapy technicians at simulation and treatment. The quality and reproducibility of breath-hold were checked by continuous cine-MR guidance during simulation and treatment. Breath-hold was usually performed in physiologic end-expiration.

### 2.4. Treatment Planning

The GTV was delineated using the data from CT and MRI. Organs at risk (OAR) were delineated on MRIdian^®^ simulation images. A 5 mm isotropic margin was added to the gross tumor volume (GTV) to obtain the planning target volume (PTV). An optimization structure named optimized PTV (PTV opt) could be used if critical digestive structures were close to the PTV. It was defined as follows: PTVopt = PTV − (digestive OAR + 5 mm). OAR dose constraints are listed in Table 1. Priority was always given to OAR dose constraints. The prescribed dose was 35, 36, 40 and 50 Gy in 3 to 5 fractions. Treatment planning was carried out using the ViewRay^®^ Treatment Planning System (TPS) (Oakwood Village, OH, USA), using a Monte Carlo algorithm, with normalization on median dose (D50%), meaning that 100% of the prescribed dose covers 50% of the target volume, trying to ensure 95% of PTVopt coverage within the 95% isodose and 99% of GTV coverage with the 99% isodose. Treatment was delivered using step-and-shoot intensity-modulated radiation therapy (IMRT) with 6-MV photons and approximately 15–20 beams and 70–90 segments. No concomitant chemotherapy was administered during radiotherapy.

### 2.5. Daily Adaptive Treatment Workflow

After daily TRUFISP image acquisition, patients were positioned to the adrenal gland target. After rigid registration of the GTV, OAR contours were propagated on the daily MR image using deformable image registration. OAR contours not considered optimal were modified by the physician (especially digestive OAR contours), but also adrenal target volume if needed). The initial plan was then evaluated by the physician and the physicist on the new contours, leading to the predicted plan. If all dose constraints were met, no adaptation was required (non-adapted fractions). If a decrease in tumor coverage and/or unacceptable OAR dose constraints were observed, the initial plan was optimized on the integrated TPS, leading to an adapted plan (and an adapted fraction). The electron density map (transferred from the CT to MR images) and the skin contour were checked to ensure correct dose recalculation [11].

Quality assurance of the newly optimized plan was performed by recalculating the plan with a secondary Monte Carlo algorithm before irradiation. Gating was ensured by following a structure with good spontaneous contrast on MRIdian^®^ (Viewray, Oakwood Village, USA) acquisition (usually the GTV itself) on sagittal images obtained by cine-MR. The beam was turned off when more than 5% of the tracked structure was outside the threshold of 3 mm from its initial position.

### 2.6. Clinical Assessment, Dosimetric Evaluation, and Evolution of GTV

The primary endpoint was the assessment of acute and late toxicities. Secondary endpoints were the impact of adaptive treatment on target volume coverage and OAR sparing as well as the local control rate and overall survival (OS).

Follow-up started on the first day of MRgRT treatment until death or latest news for each patient. All patients were regularly assessed during and after treatment. The assessment included clinical and radiological (CT scan, MRI, or PET scan) evaluations at each visit.

All toxicity events were reported according to the Common Terminology Criteria for Adverse Events (CTCAE) v5.0 at each clinical examination. Acute toxicities were defined as toxicities occurring during treatment until 3 months post-treatment. Late toxicities were defined as toxicities occurring after 3 months post-treatment.

The evaluation was conducted according to the Response Evaluation Criteria in Solid Tumors (RECIST) criteria including local complete response (CR), local partial response (PR), local stable disease (SD), local recurrence (LR) and local progressive disease (PD) based on radiological assessment.

For each adapted fraction delivery, the predicted plan (initial plan on the daily image) and the adapted/delivered plan (new plan on the daily image) were compared posteriori with the initial plan. PTV and GTV coverage (D2%, D95%, D98%, V100%, V95%, and V90%) values as well as OAR maximum dose and volumetric doses were recorded.

GTV size was measured at each fraction and compared during treatment from the first to the last fraction.

### 2.7. Statistical Analysis

All analyses were performed using SPSS version 20.0 and GraphPad PRISM version 9.4.

All events were measured from the first day of MRgRT treatment.

The survival rates were estimated using the Kaplan–Meier method.

Comparison between predicted and adapted/delivered was performed by a paired Wilcoxon test. Statistical significance was established at *p* < 0.05.

## 3. Results

### 3.1. Patient and Treatment Characteristics at Inclusion

Twelve consecutive patients treated with MRgRT between 14 November 2019 and 17 August 2021 for an adrenal gland metastasis were included. The median age of the patients was 74 years (range 51–86). There were six men (50%) and six women (50%). The primary cancer was lung cancer in 66.7% of the cases, kidney cancer in 16.7% of the cases, bladder cancer in 8.3% of the cases and prostate cancer in 8.3% of the cases. The majority of patients were oligometastatic (58.3%). Five patients had more advanced diseases (41.7%). The metastases were located in the right adrenal metastasis in 6 cases (50%), the left one in 5 cases (41.7%), and both in 1 case (8.3%). All patients had an ECOG score of 1 (100%). Previous treatments were primary cancer surgery (27%), chemotherapy (45%), and radiotherapy of the primary tumor (27%), immune checkpoint inhibitors (55%), targeted therapy (27%), and hormone therapy (9%). Systemic therapy (unless immune checkpoint inhibitors) was discontinued in all patients at the time of radiotherapy. Concomitant immune checkpoint inhibitors were administered in four cases (33.3%). Patient characteristics are summarized in Table 2.

All included patients were entirely treated and no treatment interruption was required.

### 3.2. Initial Treatment Plans

The prescribed dose was 50 Gy (BED10 = 100 Gy, 10 Gy × 5 fractions) for 6 patients, 40 Gy (BED10 = 72 Gy, 8 Gy × 5 fractions) for 1 patient, 36 Gy (BED10 = 79.2 Gy, 12 Gy × 3 fractions) for 4 patients, and 35 Gy (BED10 = 59.5 Gy, 7 Gy × 5 fractions) for one patient. The median GTV was 21.12 cc (range 6.85–62.53). The median V95% of PTV was 95.74% (76.96–98.57). The median V95% of PTVopt was 95.26% (73.76–100). Two patients did not achieve the PTV coverage objective because of the proximity of an OAR. The initial plan respected the dose constraints in all delineated OARs except for the stomach in one patient (Dmax: 34.3 Gy, V18: 3.5 cm^3^), but the treatment was accepted as the D_0.03_ cm^3^ was respected (<32 Gy). Table 3 presents the dosimetric data of initial plans.

### 3.3. Dosimetric Benefits of Adaptive MRgRT

Thirty-eight (69%) fractions were adapted because of a dosimetric benefit obtained either on PTV coverage or on OAR sparing. Seventeen (31%) fractions were not adapted and the patients were treated with the initial plan. The median treatment duration of adapted fractions was 78 min, including patient preparation, positioning, image acquisition, image registration, OAR recontouring, plan adaptation, and treatment delivery. The dosimetric data comparing predicted and adapted plans are presented in Table 4. Priority given to the protection of OARs was not to the detriment of PTV coverage. Figure 1 shows an example of the benefit of adaptation on PTV coverage for a given fraction.

PTV coverage was significantly improved for adapted plans compared to predicted plans (median PTV V95% increased from 89.85% to 91.17%, *p* = 0.0478). The plan adaptation also significantly improved Dmax for all OARs. Figure 2 and Figure 3 show the target volume and organs at risk mean dose variations between predicted and adapted fractions. The OAR’s dosimetric results for the predicted and optimized plans of five and three fractions are presented in Table 5.

### 3.4. Evolution of GTV Size during Treatment

The size of GTV treated before and during each fraction of radiotherapy remained stable for nine patients, and increased at each fraction for three patients from a median volume of 21.12 cc, 23.44 cc, and 56.18 cc before treatment to, respectively 21.83 cc, 25.49 cc, and 61.04 cc at the last fraction.

### 3.5. Evaluation of Tolerance

The most common grade 1–2 acute toxicities were asthenia (33.3%) and nausea/vomiting (16.7%). No patient suffered from grade 3 or greater toxicity. With a median follow-up of 15.5 months, no patient presented late toxicities. Table 6 describes the tolerance of treatment according to the CTCAE v5.0.

### 3.6. Clinical Results

The median follow-up was 15.5 months, (range 5–23). The 1–year LC rate was 100%.

At last follow-up, 3 patients (25%) presented a complete response (LCR), 1 patient (8.3%) a local partial response (LPR), 6 patients (50%) a local stable disease (LSD), and 2 patients (16.7%) a local recurrence (LR).

The median OS was not reached. The 1-year OS rate was 91.7%. The mean DFS was 20.1 months (95% CI 12.6–20.4). The 1–year DFS rate was 91.7%.

At the last follow-up, 1 patient (8.3%) died and 11 (91.7%) were still alive. A total of 9 (75%) patients experienced recurrences, including two patients (16.7%) with a local recurrence evaluated by a CT scan. The two patients had lung cancer as primary cancer. The first patient developed a local recurrence 13 months from the end of radiotherapy and the second case 23 months after the end of radiotherapy. Figure 4 shows the LC and OS curves.

## 4. Discussion

To the best of our knowledge, the current study is the first to evaluate toxicities and outcomes in patients treated with MRgRT for and adrenal gland metastasis (AGM). We report here our experience with the first twelve patients treated in our institution with a median follow-up of 15.5 months.

Stereotactic body radiotherapy (SBRT) treatment of adrenal gland metastases is a good alternative to surgery. The proximity of the abdominal organs requires optimal treatment based on reliable image-guided radiotherapy (IGRT) to deliver high-precision treatment. MRgRT offers the possibility to perform adaptive radiotherapy thanks to integrative artificial intelligence-based software. It allows for monitoring the anatomical variations on a daily basis and adopting the dosimetric plan. Online re-planning (with the patient on the treatment table) is possible using a daily redelineation, plan evaluation and dosimetric optimization in order to take into account anatomic variations [17,18]. Furthermore, MRIdian^®^ allows for continuous gating with cine-MR acquisitions.

In our study, the treatment was well tolerated. No toxicity of grade 3 or higher was reported. Consistent with results from other SBRT for AGM studies, with follow-up varying from 7 to 41 months [8,9], the most frequent grade 1 and 2 toxicities were mainly nausea and fatigue with an overall incidence of 30–50% (versus 33.3% in our study) [19,20].

A recent pooled meta-analysis and systematic review of 39 studies of SBRT for AGM involving 1006 patients reported an overall rate of grade 3 or higher toxicity of 1.8%. Fifteen patients experienced grade 3 toxicities. These included nausea (*n* = 3), diarrhea (*n* = 1), gastrointestinal bleeding (*n* = 1), esophageal ulcer (*n* = 1), gastric/duodenal ulcers responding to medical management (*n* = 2), hypertensive emergency (*n* = 1), and unknown (*n* = 6). Two patients (0.2%) experienced CTCAE grade 4 toxicity (gastrointestinal bleeding from a duodenal ulcer, and a perforated pyloric ulcer 14 months after radiation). It should be noted that this meta-analysis reported a single possible CTCAE grade 5 toxicity in a patient receiving nivolumab with SBRT who developed an immune reaction to nivolumab after SBRT, with severe abdominal pain and diarrhea [21].

In the current study, one patient received nivolumab with MRgRT, without adverse events during and after treatment.

Other authors evaluated MRgRT for AGM, but they focalized their attention on dosimetric benefits.

In a recent study, initial plans of VMAT CT-based image-guided radiotherapy (CT-IGRT) with breath-holding (BH) showed superior results to the initial plan of IMRT MRgRT-BH in terms of improvements in target coverage (*p* = 0.02), conformality, and D0.5 cc to the large bowel, duodenum, and mean ipsilateral kidney. However, the importance of interfractional target changes, plan adaptation to the anatomy of the day, and real-time motion tracking has granted MRgRT-BH the ability to safely provide ablative doses to adrenal lesions near mobile luminal OAR. Finally, non-adaptive CT-IGRT-BH had a 71.8% frequency of predicted indications for adaptative treatment [22].

The benefit of daily adaptation was demonstrated by Palacios et al. in 2018 in a series of 17 patients treated for adrenal gland metastasis in 84 fractions. They concluded that online re-optimization of treatment plans led to significant improvements in target coverage and OAR sparing, which is in accordance with our results.

The dosimetric benefit of adapted MRgRT has been demonstrated for different clinical indications [23,24]. Acharya et al. published 2015 their experience with the first clinically implemented online adaptive MRgRT system for patients with abdominopelvic cancers. The initial cohort included 5 patients, then they extended their cohort to 20 patients (170 fractions). A re-optimization was performed for 30.6% of patients and 54.4% were treated with a previously adopted plan. In our study, 38 (69%) fractions were adapted because of a dosimetric benefit obtained either on PTV coverage or on OAR sparing. Seventeen (31%) fractions were not adapted and patients were treated with the initial plan.

De Kost et al. evaluated the changes in stomach volume for the treatment of left AGM. They performed daily MR scans pre- and post-treatment at each fraction. The analysis revealed frequent decreases in stomach volumes and in the predicted stomach doses. In our study, our patients could be asked to fast for at least 3 to 4 h prior to all simulation exams and every fraction in order to maintain reproducibility and to minimize stomach movement, in case of close stomach or duodenum proximity. Indeed, the results of this study support a role for dietary instructions to enable better OAR sparing for left adrenal SBRT [25].

The normal tissue complication probability (NTCP) framework analysis revealed that patients with left adrenal tumors were more likely to benefit from MR-guided daily on-table adaptive SBRT using current dose/fractionation regimens. This is due to reductions in predicted gastric toxicity resulting from their anatomical proximity to the stomach, a mobile and dose-sensitive organ. On the other hand, right-sided adrenal lesions may be considered for dose escalation due to low predicted NTCP [26]. Daily adaptation enables a reduction in the dose received by nearby OARs and an increase in the coverage of target volumes. The median time for re-contouring, re-optimization, and quality assurance was only 26 min. The same team published in 2017 the results of 2 years of experience with the MRgRT in about 316 patients of which 50% had an abdominopelvic tumor (5 adrenal sites (7%)) and thirty (30%) were treated with SBRT. They evaluated the clinical feasibility of online adaptive MR-guided SBRT for oligometastatic and unresectable primary malignancies of the abdomen and central thorax. The incidence of plan adaptation of the SBRT treatment was 84% (81 of 97 fractions) and the average treatment time was 28 min. They recommended using this technique for hypofractionated treatments (less than 10 fractions) [27].

A recently published German retrospective study demonstrated the benefit of adaptive MRgRT treatment for various cancers in terms of improved PTV coverage for all subgroups. The largest median improvements in GTV near-minimum dose (D98%) were found for the liver (6.3%, *p* < 0.001), lung (3.9%, *p* < 0.001), and abdominal lymph nodes (6.8%, *p* < 0.001) subgroups [28].

We recently evaluated adaptive MRgRT for pancreatic tumors and showed that the systematic daily adaptation of the treatment plan brought a statistically significant benefit for the stomach, the duodenum, and the jejunum, along with a significant improvement in PTV coverage V95% (3,19% *p* < 0.003), V80% (2,04% *p* < 0.0015), and GTV V95% coverage (1,06% *p* < 0.04974). It improved the overall dosimetric quality of MRI-guided radiotherapy, which are roughly the same findings as in the current study [16].

In our study, 3 patients experienced a GTV increase between fractions, reinforcing the interest in daily adaptive radiotherapy with daily delineation of GTV if needed.

In our series, 8 patients were treated with five fractions, applying a median BED10 of 100 Gy. The 1-year local control, 1-year disease-free survival and 1-year overall survival were, respectively 100%, 91.7%, and 91.7%. These findings are comparable to the results of a recent pooled meta-analysis and systematic review of 39 studies of SBRT for AGM. Chen et al. revealed a dose–response between the reported median or mean BED10 and 1 and 2-year LC and a weaker but significant association between BED10 and 2 years OS. Based on a meta-regression of BED and LC, a BED10 of 60 Gy, 80 Gy and 100 Gy predicted 1-year LC of 70.5, 84.8 and 92.9% and 2- year LC of 47.8, 70.1 and 85.6%, respectively [21].

In the study of Salama et al. evaluating SBRT for the treatment of adrenal gland metastases, the 1-year LC rate was 73%, the 1-year distant control was 30%, and the 1-year OS was 90%. All treated patients were in an oligometastatic state [20]. Van Vliet et al. reported treatment patterns of patients treated for AGM using surgery or SBRT. After a median follow-up of 21 months in the surgery cohort and 16.9 months in the SBRT cohort, they noted a local control rate of 74% for surgery and 96% for SBRT (*p* = 0.003) [29].

There are several limitations to our study. First, this is a single-center study with limited accrual, and our results need to be compared with other teams using MRgRT. However, we believe that these preliminary results support the interest in continuing the evaluation of this technique in this indication. Secondly, the main limitation of this technique is the treatment duration (median fraction duration in our study was 78 min), requiring therefore an optimal selection of patients. Moreover, we described in our series an excellent local control rate, but frequent distant relapse. Finally, our follow-up is short and a longer follow-up is required to assess late toxicities and clinical results.

## 5. Conclusions

MRgRT for adrenal gland metastases is safe with good clinical results in terms of local control. Daily adaptation of the treatment plans improved the overall dosimetric quality of MRI-guided radiotherapy with better target volumes coverage and protection of OARs. A longer follow-up is required to assess late toxicities and clinical results.

## Figures and Tables

**Figure 1 jcm-12-00291-f001:**
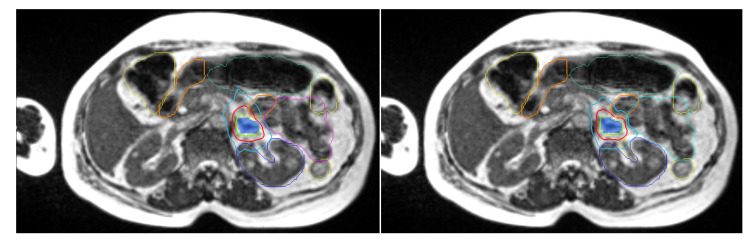
Comparison of dose distribution between predicted (**left**), and adapted/delivered (**right**) dosimetry on MR 0.35-T TRUFISP images for a prescription of 50 Gy in 5 fractions. Isodose line, 47.5 Gy in green, 32 Gy in red, and 20 Gy in cyan. Duodenum in orange, PTV in blue colorwash.

**Figure 2 jcm-12-00291-f002:**
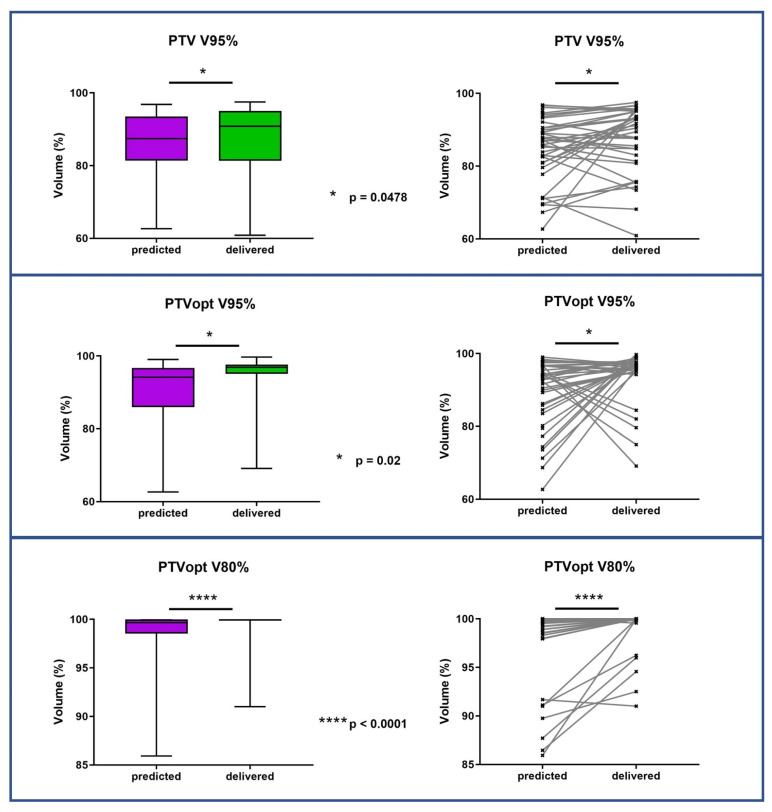
PTV and PTVopt coverage variations between predicted fractions (baseline plan on daily anatomy) and delivered fractions (new plan on daily anatomy). PTV = planning target volume; PTVopt = PTV optimized.

**Figure 3 jcm-12-00291-f003:**
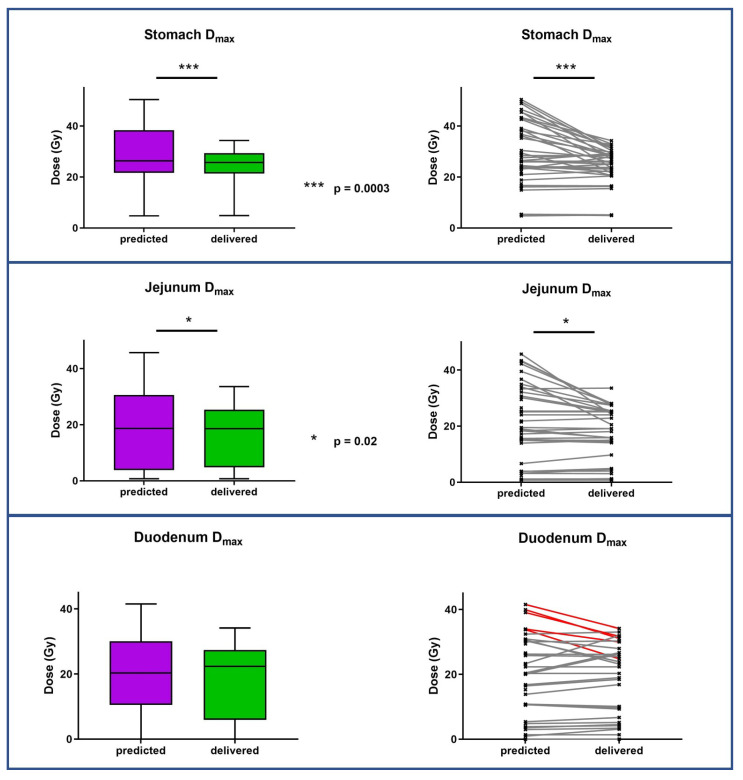
Organs at risk dose variations between predicted fractions (baseline plan on daily anatomy) and delivered fractions (new plan on daily anatomy).

**Figure 4 jcm-12-00291-f004:**
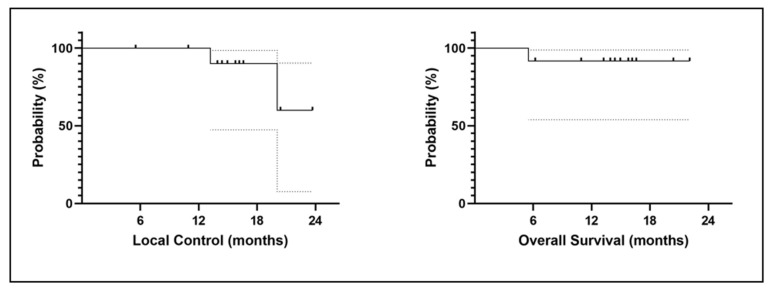
Local control (**left**) and overall survival (**right**) curves.

**Table 1 jcm-12-00291-t001:** Organ at risk dose constraints.

Organ	Dose Constraints (5 Fractions)	Dose Contraints (3 Fractions)
**Stomach**	Dmax < 32 Gy V18 Gy < 10 cm^3^	Dmax < 24 Gy V 16 Gy< 10 cc
**Duodenum**	Dmax <32 Gy V18 Gy < 5 cm^3^	Dmax < 24 Gy V 15 Gy< 5 cc
**Small intestine**	Dmax < 32 Gy V 19.5 < 5 cm^3^	Dmax < 27 Gy V 16 Gy < 5 cc
**Large intestine**	Dmax < 32 Gy V 25 Gy < 5 cm^3^	Dmax < 30 Gy V 20 Gy < 20 cc
**Kydneys**		V 14.4 Gy < 178 cc
**Kidneyipsilateral**	V18 Gy < 33% V < 14.5 Gy > 130 cm^3^ (if single kidney)	V 12.1 Gy < 59 cc
**Liver**	V 15 Gy > 700 cm^3^	Dmoy < 15 Gy V 15 Gy < 700 cc
**Spinal cord**	Dmax < 25 Gy	Dmax < 22 cc

Gy = Gray.

**Table 2 jcm-12-00291-t002:** Patient characteristics.

Gender	
**Women** **Men**	6 (50%) 6 (50%)
**Median age (range)**	74 years (51–86)
**Primary tumor**	
Lung Kidney Bladder Prostate	8 (66,7%) 2 (16,7%) 1 (8.3%) 1 (8.3%)
**Stage**	
Oligometastatic Multimetastatic	7 (58.3%) 5 (41.7%)
**Side**	
Right adrenal Left adrenal Bilateral	6 (50%) 5 (41,7%) 1 (8.3%)
**WHO score**	
0 1	0 12 (100%)
**Previous treatment received**	
Primary cancer surgery Chemotherapy Radiotherapy of the primary tumor Immune Checkpoint inhibitor or targeted therapy Hormone therapy	2 (22%) 5 (56%) 2 (22%) 5 (56%) 1 (11%)
**Concomitant treatment**	
Chemotherapy Hormone therapy Immune Checkpoint inhibitor-Pembrolizumab -Nivolumab	0% 0% 4 (33.3%) 3 (25%) 1 (8.3%)
**Total dose**	45 Gy (35–50)
**Median dose per fraction**	10 Gy (7–12)
**MedianTreatment duration**	7 days (5–10)
**Median Fraction duration**	78 min (45–170)
**Median follow up**	15.5 months (5–23)
**Evolution (1 year)**	
Local complete response Local partial response Local Stable disease Local recurrence Metastatic progression Death	1 (8.3%) 1 (8.3%) 2 (16.7%) 2(16.7%) 6 (50%) 1 (8.3%)

WHO = World Health Organization.

**Table 3 jcm-12-00291-t003:** Target volume and OAR’s dosimetric results for the initial plan.

	Target Volume Median (Min–Max)	OAR (3 Fractions) Median (Min–Max)	OAR (5 Fraction) Median (Min–Max)
**PTV-opti** V100% V95% V80%	67.7% (47.41–91.55) 95.26% (73.76–100) 99.27% (94.11–100)		
**PTV** V100% V95% V80%	64.84% (47.41–85.57) 95.74% (76.96–98.57) 99.44% (90.07–100)		
**GTV** V100% V95% V80%	81.16 (53.49–97.44) 99.90 (77.98–100) 100 (90.24–100)		
**Kidney**		**V12.1 Gy:** 10.6 (1.13–52.59)	**V18 Gy:** 4.69 cc (0–49.56)
**Spinal cord**		**Dmax:** 10.685 (9.63–11.65)	**Dmax:** 12.55 Gy (6.47–20.74)
**Stomach**		**Dmax:** 20.79 (7.49–22.16) **V16 Gy:** 2.12 (0–2.2)	**Dmax:** 28.54 Gy (4.06–34.37) **V18 Gy:** 3.89 cc (0–7.72)
**Duodenum**		**Dmax:** 12.28 (0.62–23.68) **V15 Gy:** 0 (0–3.96)	**Dmax:** 26.28 Gy (2.12–30.53) **V18 Gy:** 1.06 cc (0–4.4)
**Small intestine**		**Dmax:** 7.49 (0.85–18.32) **V16 Gy:** 0 (0–0.82)	**Dmax:** 26.04 Gy (4.35–30.74) **V19.5 Gy:** 0.79 cc (0–3.86)
**Large intestine**		**Dmax:** 0.55 **V20 Gy:** 0	**Dmax:** 19.75 Gy (0.63–29.02) **V25 Gy:** 0 cc (0–0.45)

OAR = organ at risk; PTV = planning target volume; GTV = gross tumor volume.

**Table 4 jcm-12-00291-t004:** Average target volume dosimetric results for the predicted and adapted/delivered plans.

Target Volume	Predicted Plan Median (Min–Max)	Adapted/Delivered Plan Median (Min–Max)	*p*-Value
**PTV-opti** V100% V95% V80%	75.69% (11.89–92.7) 94.91% (62.68–99.02) 99.83% (85.93–100)	67.31% (43.99–92.12) 96.68%(69.11–99.71) 99.97% (91.01–100)	0.44 **0.02** **<0.0001**
**PTV** V100% V95% V80%	73.95% (11.51–88.22) 89.85% (62.68–96.82) 97.1% (84.92–100)	63.75% (38.63–88.36) 91.17% (60.88–97.53) 96.1% (81.47–100)	0.58 **0.0478** 0.68
**GTV** V100% V95% V80%	87.58% (15.07–99.35) 96.86% (79.67–100) 99.24% (90.55–100)	76.83% (51–96.75) 96.44% (74.87–100) 98.46% (90.44–100)	0.72 0.20 0.34

Differences are statistically significant in case of *p*-value < 0.05 (value in bold).

**Table 5 jcm-12-00291-t005:** OAR’s dosimetric results for the predicted and adapted/delivered plans.

OAR	Predicted Fractions Median (Min–Max)	Adapted/Delivered Fractions Median (Min–Max)	*p*-Value
**Kidney 5 fractions** V18 Gy **Kidney 3 fractions** V12.1 Gy	2.98 cc (0–48.85) 31.76 cc (13.08–55,07)	3.06 cc (0–49.1) 32.92 cc (14.2–62.16)	0.20
**Spinal cord 5 fractions** Dmax **Spinal cord 3 fractions** Dmax	12.28 Gy (5.08–20.97) 11.1 Gy (8.87–13.05)	12.15 Gy (0.21–21.77) 10.9 Gy (8.86–12.74)	0.93
**Stomach 5 fractions** Dmax **Stomach 3 fractions** Dmax	28.41 Gy (14.86–50.38) 16,76 Gy (4.75–38.53)	26.98 Gy (15.47–34.28) 17.89 Gy (4.86–23.7)	**0.0003**
**Duodenum 5 fractions** Dmax **Duodenum 3 fractions** Dmax	19 Gy (1.8–36.3) 15.79 Gy (0–33.8)	19.8 Gy (0.1–32) 13.29 Gy (0–24.81)	0.64
**Small intestine 5 fractions** Dmax **Small intestine 3 fractions** Dmax	21.47 Gy (2.35–45.69) 9.58 Gy (0.76–19.54)	15.88 Gy (3.07–57.17) 9.21 Gy (0.76–19.09)	**0.02**
**Large intestine 5 fractions** Dmax **Large intestine 3 fractions** Dmax	18.34 Gy (4.98–26.55) 1.48 Gy (0–6.51)	18.23 Gy (0–29.54) 1.38 Gy (0–6.89)	0.73

OAR = organ at risk. Differences are statistically significant in case of *p*-value < 0.05 (value in bold).

**Table 6 jcm-12-00291-t006:** Assessment of tolerance.

CTCAE v5.0	Toxicity (0–90 Days)	Toxicity (6 Months)	Toxicity (9 Months)	Toxicity (1 Year)
Abdominal pain g0 g1 g2 g3 Ongoing	12 (100%) 0 0 0 0	12 (100%) 0 0 0 0	12 (100%) 0 0 0 0	12 (100%) 0 0 0 0
Nausea/vomiting g0 g1 g2 g3 Ongoing	10 (83.4%) 1 (8.3%) 1 (8.3%) 0 0	12 (100%) 0 0 0 0	12 (100%) 0 0 0 0	12 (100%) 0 0 0 0
Gastritis/Enteritis g0 g1 g2 g3 Ongoing	12 (100%) 0 0 0 0	12 (100%) 0 0 0 0	12 (100%) 0 0 0 0	12 (100%) 0 0 0 0
Peptic ulcer g0 g1 g2 g3 Ongoing	12 (100%) 0 0 0 0	12 (100%) 0 0 0 0	12 (100%) 0 0 0 0	12 (100%) 0 0 0 0
Digestive fistula g0 g1 g2 g3 Ongoing	12 (100%) 0 0 0 0	12 (100%) 0 0 0 0	12 (100%) 0 0 0 0	12 (100%) 0 0 0 0
Diarrhea g0 g1 g2 g3 Ongoing	12 (100%) 0 0 0 0	12 (100%) 0 0 0 0	12 (100%) 0 0 0 0	12 (100%) 0 0 0 0
Asthenia g0 g1 g2 g3 Ongoing	8 (66.7%) 3 (25%) 1 (8.3%) 0 0	11 (91.7%) 1 (8.3%) 0 0 0	12 (100%) 0 0 0 0	12 (100%) 0 0 0 0

CTCAE = Common Terminology Criteria for Adverse Events.

## Data Availability

Data available upon request.

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
