# Peer review of "Stereotactic MR-Guided Radiotherapy for Adrenal Gland Metastases: First Clinical Results"

_jcm, 2022, doi:10.3390/jcm12010291_

Round 1

Reviewer 1 Report

This is a study of 12 patients treated with gated MR-guided adaptive radiotherapy. This is a small number of heterogeneous patients but the manuscript is generally well written and the conclusions reasonable.

General comments

• On should always be prudent in claiming to be the “first”, are these not series of similar treatments?

https://doi.org/10.1016/j.radonc.2021.11.024

https://www.sciencedirect.com/science/article/pii/S0167814021090204

• There are many typos in the punctuation which should be reviewed.

• There is no discrimination made between MR-guidance and adaptive radiotherapy.

Introduction

• There references in the first paragraph are not epidemiological references on the incidence of adrenal metastases.

• With the availably randomized data for radiotherapy of oligometastases, especially in lung cancer, it is arguable that SBRT is the standard and not surgery.

Methods

• What is PTVopt?

• The prescription dose appears somewhat arbitrary as it does not follow ICRU guidelines and the target coverage is variable.

• Was this a prospective clinical trial? A registry? Were patients treated for adrenal metastases outside of this trial?

• When were patients censured for local control? At last imaging? When starting a new systemic treatment?

• Were antibodies or hormone therapy also held during treatment (or only TKIs and cytotoxics)?

• Were patients given proton pump inhibitors?

• Please clarify normalization of dose to D50..

• It would probably be preferable to use a term other than “tracking” to avoid confusion.

Results

• When looking at local control, how many patients had SBRT + another systemic treatment?

• Where any images taken at the end of treatment, how do we know what the actual dose delivered may have been?

Discussion

• How much of the benefit of adaptation could have been had by simply optimizing a 6D match (optimizing a 6D match dosimetrically vs. doing a 3D GTV match)?

• What would the expected benefit be over gated treatment with a faster device using adaptative CT-based treatment?

• The quoted 40% GI toxicity for “conventional” SBRT is not representative (nor the 80% vs 33.3% grade 1-2 toxicities).

Reviewer 2 Report

The paper reports toxicities and outcomes in patients treated with Magnetic resonance-guided radiotherapy MRgRT for and adrenal gland metastasis. Different prescriptions are considered. 12 patients are analyzed. The number of patients is limited and there is differences among the used fractionation.

It might be interesting to correlate the toxicities with the delivered treatments.

The paper is interesting from the clinical point of view.
